# Short-Chain Fatty Acids in Gut–Heart Axis: Their Role in the Pathology of Heart Failure

**DOI:** 10.3390/jpm12111805

**Published:** 2022-11-01

**Authors:** Midori Yukino-Iwashita, Yuji Nagatomo, Akane Kawai, Akira Taruoka, Yusuke Yumita, Kazuki Kagami, Risako Yasuda, Takumi Toya, Yukinori Ikegami, Nobuyuki Masaki, Yasuo Ido, Takeshi Adachi

**Affiliations:** Department of Cardiology, National Defense Medical College, 3-2 Namiki, Tokorozawa 359-8513, Japan

**Keywords:** short-chain fatty acid, acetate, butyrate, propionate, histone deacetylases, G-protein-coupled receptors, heart failure

## Abstract

Heart failure (HF) is a syndrome with global clinical and socioeconomic burden worldwide owing to its poor prognosis. Accumulating evidence has implicated the possible contribution of gut microbiota-derived metabolites, short-chain fatty acids (SCFAs), on the pathology of a variety of diseases. The changes of SCFA concentration were reported to be observed in various cardiovascular diseases including HF in experimental animals and humans. HF causes hypoperfusion and/or congestion in the gut, which may lead to lowered production of SCFAs, possibly through the pathological changes of the gut microenvironment including microbiota composition. Recent studies suggest that SCFAs may play a significant role in the pathology of HF, possibly through an agonistic effect on G-protein-coupled receptors, histone deacetylases (HDACs) inhibition, restoration of mitochondrial function, amelioration of cardiac inflammatory response, its utilization as an energy source, and remote effect attributable to a protective effect on the other organs. Collectively, in the pathology of HF, SCFAs might play a significant role as a key mediator in the gut–heart axis. However, these possible mechanisms have not been entirely clarified and need further investigation.

## 1. Introduction

Heart failure (HF) is a global clinical and socioeconomic burden in developed countries worldwide. Despite the development of novel diagnostic and therapeutic modalities, its prognosis remains poor.

There is accumulating evidence implicating the gut microbiota as critical contributors to host health and homeostasis [1,2]. There has also been increasing evidence supporting the relationship between the gut microbiota and HF [2]. In the setting of HF, the microenvironment of the gut (e.g., hypoxia, hypercapnia, changes in local pH, and redox state) can be altered, and these are all known to be potent activators of bacterial virulence in the microbiota [3]. Thus, the composition of intestinal microbiota may shift rapidly during intestinal ischemia and reperfusion [4,5], or following an increase in portal vein pressure [6]. Although clinical evidence is limited, intestinal hypoperfusion owing to reduced cardiac output and congestion due to increased venous pressure can further disrupt the barrier function of the intestine and promote systemic inflammation through bacterial translocation, potentially leading to further HF exacerbations.

The contribution of gut microbiota on the pathology of HF may be achieved, at least in part, through the gut microbiota-derived substance, short-chain fatty acids (SCFAs) [7]. SCFAs are fatty acids with a carbon chain of up to six atoms, which consist of acetate (C2), propionate (C3), and butyrate (C4). SCFAs are produced through the fermentation of dietary fiber and resistant starches by specific gut anaerobic bacteria [8,9]. The process involved in the production of SCFAs from fiber involves complex enzymatic pathways that are active in an extensive number of bacterial species. The molar ratios of SCFA in human fecal content are 50–60, 15–20, and 10–20 for acetate, propionate, and butyrate, respectively [10,11]. The proportions of acetate, propionate, or butyrate were reported not to differ by age or sex, [12] but they can vary depending on factors such as diet, microbiota composition, site of fermentation, and host genotype [13]. It has been estimated that SCFA metabolism consists of ~10% of our daily calorific requirements in healthy subjects [14]. 

Butyrate shows greater uptake in the colonic epithelium [11] and is the main energy source for epithelial cells in colon-regulating epithelial cell function. It also has an immunomodulatory impact on host innate and adaptive immunity [14]. Propionate shows greater uptake by the liver [11] and is utilized as a substrate of gluconeogenesis in the organ. Acetate is not absorbed or metabolized in these organs and delivered to the peripheral [14]. Although the evidence on the mechanistic link between SCFAs and the pathology of HF is still limited, SCFAs may have the potential to play a significant role in the pathology of a variety of cardiovascular diseases, including HF. In this review, we will summarize the current knowledge—most of which is based on the basic research—on the molecular action of SCFAs, the mutual interaction between SCFAs and the pathology of HF, and the potential mechanisms of SCFAs’ effect, and we will discuss the role of SCFAs as potential therapeutic targets.

## 2. The Molecular Action of SCFA

### 2.1. HDAC Inhibition by SCFA

The acetylation status of histones regulates the access of transcription factors to DNA and influences levels of gene expression. The histone deacetylases (HDACs) have been implicated in the repression of gene expression by facilitating chromatin condensation [15]. Pharmacological inhibition of HDACs activity leads to the inhibition of cell proliferation and growth. This epigenetic mechanism is a powerful regulator of tumor responses to chemotherapy and adaptation to environmental triggers (e.g., hypoxia). For decades, HDAC inhibitor was believed to have an anti-tumor effect, and the utility of this class of drugs has been explored in a variety of malignancies [16]. Some of them have already been applied for humans to treat malignancies in clinical practice.

Recent studies have shown that HDAC activities also participate in regulating the hypertrophic response of the heart [15,17,18]. Specifically, experimental animals with pressure-overload-induced cardiac hypertrophy or myocardial infarction exhibited the histone acetylation of genes associated with cardiac contraction, collagen deposition, inflammation and extracellular matrix, as well as the activation of NF-κB target genes. HDAC inhibition has been shown to blunt these pathological changes in vitro and in vivo [19,20,21,22,23]. This effect was mediated by the suppression of autophagy [22]. HDAC inhibition promotes the acetylation of non-histone proteins such as the mineralocorticoid receptor, which might mediate the attenuation of cardiac hypertrophy and fibrosis in spontaneously hypertensive rats [24]. HDAC inhibition was also shown to protect the heart from ischemic injury. HDAC inhibitor reduced myocardial infarct size in an experimental animal model, even when delivered in the clinically relevant context of reperfusion [25].

HDAC activity was shown to be inhibited in high-fiber fed mice [26]. This phenomenon might arise from the production of SCFAs. SCFAs show the inhibitory effect of HDACs accompanied by the inhibition of NF-κB activity, which is particularly characteristic of butyrate and propionate [27,28]. HDAC inhibition by SCFAs depends not only on the type of SCFA, but also on the organs or cell types on which they act [26]. Butyrate and propionate were shown to reduce NF-κB activity in PBMCs in an analogous manner to a typical HDAC inhibitor, trichostatin A [29].

### 2.2. SCFAs’ Effect via G-Protein-Coupled Receptors (GPRs)

To date, four free fatty acid receptors (FFARs) have received considerable attention owing to their physiological importance in various biological processes. Of the FFARs identified, GPR40 (FFAR1) and GPR120 (FFAR4) are activated by middle-chain FA (GPR40) and long-chain FA (GPR40 and GPR120), while GPR43 (FFAR2), GPR41 (FFAR3), and olfactory receptor 78 (Olfr 78) are activated by SCFAs [30,31]. GPR41 and GPR43 used to be deemed as orphan G-protein coupled receptors that vary in specificity for individual SCFAs, intracellular signaling, and tissue localization [32]. GPR43 exhibits comparable agonist activities for acetate, propionate, and butyrate, whereas GPR41 displays remarkable differences in the receptor affinity (propionate ≥ butyrate > acetate) [32]. GPR109A is a receptor for niacin and mediates the lipid-lowering actions of the vitamin [33]. GPR109A is expressed in the intestinal epithelial cells and recognizes butyrate as a ligand [34]. Olfr 78 is an olfactory receptor, which was identified in the other organs [35]. 

Like the other GPRs, these receptors share a 7-transmembrane helix framework and are localized at the cell membrane. When agonists bind to these receptors, they activate heterotrimeric G-proteins on the intracellular side, which synthesize the second messengers such as cyclic 3′,5′-adenosine monophosphate (cAMP) or phospholipase C-β, leading to the activation of downstream signal pathways exerting a variety of effects [7,36]. Only a small portion of these pathways have been identified and rigorously investigated by many research groups.

GPRs can influence many physiological functions via signaling pathways and self-modulation. FFARs are widely expressed in various tissues and contribute to many important physiological functions that maintain energy and immune homeostasis [7]. GPRs’ signaling should be protective against HF, since GPR41-, GPR43-, and GPR 109A-knockout mice fed with normal chow showed cardiac hypertrophy, elevated left-ventricular end-diastolic pressure, and perivascular fibrosis [37]. In particular, these pathological changes were evident in GPR43- and GPR109A-knockout mice. GPR41-KO mice was also shown to have hypertension, a thickened aortic wall, and increased pulse wave velocity, indicating vascular remodeling [38]. GPRs are known to be expressed in various organs [32,38,39]. However, the gene expression of GPR41, GPR43, and GPR109A is low in the kidney, arteries, and heart, and higher in immune systems (e.g., spleen, appendix, bone marrow) and in the small and large intestine, [37,40]. One study reported that GPR43 or Olfr78 were not expressed in the kidney or heart of mice [41]. Given the well-established role of these receptors in immune function [42] and high expression in immune cells, the actions of GPR43/109A on cardiac structure or performance might be, at least partially, mediated by the changes in immune systems.

## 3. The Change of SCFA and Related Signals in Cardiovascular Diseases including HF

The changes of SCFA concentration were reported to be observed in various cardiovascular diseases including HF in experimental animals and humans [41,43,44,45]. The involvement of dysbiosis concomitant with cardiovascular disease including HF [23,46,47,48,49] may, at least in part, play a role in these phenomena.

Plasma butyrate concentration was reported to be significantly lower in hypertensive subjects compared to healthy controls [43]. This might be due to, at least in part, a lowered abundance of SCFA-producing bacteria [41,43,49,50]. Plasma butyrate level decreased in transverse aortic constriction (TAC) mice compared with the sham group. This change was accompanied by altered microbiota composition and disrupted gut epithelium, which suggests that decreased butyrate was attributable to these pathological changes and vice versa [51]. SCFA levels significantly decreased in patients with chronic kidney disease (CKD) induced by a renal ischemia reperfusion rat model; meanwhile, butyrate supplementation produced potential effects in delaying the progression of CKD. SCFA concentration significantly increased after kidney transplantation. SCFA or butyrate concentration was inversely correlated with serum creatinine concentration after kidney transplant [44].

On the other hand, however, a more recent study showed that hypertensive subjects demonstrated a rather higher plasma SCFA concentration compared to normotensive subjects. However, hypertensive subjects had lower levels of GPR43 mRNA expression in white blood cells. Interestingly, the concentration of plasma butyrate was positively correlated with blood pressure [52]. GPR43 mRNA expression was shown to be downregulated in the gut with deoxycorticosterone acetate (DOCA)-treated mice [41]. These findings suggest that the SCFA-GPR mediated blood pressure-lowering response might be blunted, but further investigation is warranted to elucidate its precise mechanism.

GPR expression might be also regulated by SCFAs. Akt2-knockout cardiomyocytes showed downregulation of GPR41, which was restored by propionate supplementation. On the other hand, Akt2-knockouts significantly upregulated the level of GPR43, the effect of which was unaffected by propionate. These findings suggested that propionate regulates the expression of GPRs, and this plays a role, at least in part, in propionate-offered beneficial effects in cardiomyocytes [53]. 

Taken together, these data suggest the possibility that cardiovascular diseases can make alterations in SCFA level and/or its related signals as a result of dysbiosis and/or the detrimental changes of the intestinal microenvironment. Especially in the case of HF, induced intestinal ischemia and/or congestion can be proposed as possible contributors [2]. Further, since SCFA has a protective effect on the cardiac function, as shown in the following sections, decreased SCFA level further can deteriorate the pathology of HF, which forms a vicious cycle (Figure 1).

To our knowledge, there has been no evidence supporting the restoration of SCFA concentration by inotropes, left-ventricular assist devices (LVAD), or heart transplantation (HT), all of which can improve cardiac output and intestinal hemodynamics. A recent study demonstrated that, contrary to expectations, microbiota-derived metabolite trimethylamine N-oxide (TMAO), which is elevated in HF and linked to poor prognosis, showed an early decrease following LVAD or HT but re-increased slowly to pre-intervention levels [54]. Lipopolysaccharide, a biomarker of endotoxemia, was also shown to remain elevated even after LVAD or HT [55]. Thus, it is hard to anticipate how the levels of SCFAs will change after improved intestinal hemodynamics by LVAD or HT, and it needs to be investigated in future.

## 4. The Potential Mechanisms of SCFAs’ Action in the Pathology of HF

SCFAs have been shown to have a multifaceted effect which directly or indirectly affects the pathology of HF (Figure 1). This includes the suppression of reactive oxygen species, restoration of mitochondrial function, lowering blood pressure, amelioration of cardiac inflammation, their utilization as an energy source, and their effect on the other organs such as kidneys, adipose tissue, and gut, most of which are considered to be beneficial. Precise molecular mechanisms mediating SCFAs’ action such as HDAC inhibition or GPR signaling are only partially identified and further investigation is warranted.

### 4.1. SCFAs Restore Mitochondrial Function

SCFAs have a protective effect on the mitochondria in various tissues. For example, colonocytes of germ-free mice deficient in SCFAs are highly energy deprived, since they show a decreased expression of key enzymes involved in fatty acid metabolism in mitochondria [10]. Consequently, these cells have a marked deficit of mitochondrial respiration as well as of oxidative phosphorylation, which can lead to autophagy. The addition of butyrate to colonocytes isolated from germ-free mice normalized this deficit [10]. 

Butyrate may promote the recovery of mitochondrial function and energy production in macrophages, which are associated with the promoted polarization to anti-inflammatory M2 macrophages rather than pro-inflammatory M1 and suppressed rat myocardial fibroblasts activity in a TAC model [51]. In cardiomyocytes, propionate attenuated the loss of mitochondrial membrane potential and contractile dysfunction induced by Akt2-knockout [53]. Butyroyloxymethyl diethylphosphate (AN-7), a pro-drug of butyrate, was shown to protect the heart and cardiomyocytes against ischemic injury in vitro and ex vivo [56], possibly through activation of the Akt and ERK survival pathway [57]. Interestingly, AN-7 diminished the hypoxia-induced dissipation of mitochondria membrane potential and cell death in cardiomyocytes, [57] but conversely, it increased cell injury and death in hypoxic cardio-fibroblasts in vitro [57]. Collectively, AN-7 showed a differential effect on hypoxia-induced mitochondrial damage depending on the cell types. AN-7 also showed this protective effect on doxorubicin (DOXO)-induced mitochondrial damage in cardiomyocytes [58]. The novel butyrate derivative phenylalanine-butyramide also protects against DOXO-induced cardiotoxicity [59], in which dysbiosis may play a role in disease progression [60]. This was accompanied by the prevention of mitochondrial reactive oxygen species release and mitochondrial dysfunction induced by DOXO [59]. 

### 4.2. SCFAs Regulate Blood Pressure

Blood pressure is closely related to the pathology of HF. In particular, the contribution of blood pressure to the onset and progression of obesity-related HF is considered to be highly significant, and its evaluation and control are the cornerstone of the clinical management of HF (see Section 4.6) [61]. The consumption of fruit and vegetables that are rich in dietary fiber is protective against hypertension [62]. Two meta-analyses found that interventions aimed at increasing total dietary fiber intake significantly reduced systolic and diastolic BPs in hypertensive patients [63,64]. Although the mechanism by which dietary fiber lowers BP still remains elusive, SCFAs may play a role in this beneficial effect.

SCFAs had vasodilator effects in isolated rat [65,66] and human arteries in vitro [67]. In angiotensin II-induced hypertensive mice, all SCFAs, acetate, butyrate, and propionate supplementation prevented BP elevation [37,43]. Among them, acetate had the most potent effect, such as lowering blood pressure and the prevention of cardiac hypertrophy [37]. Acetate significantly lowered the urine-excreted sodium-to-potassium ratio and systemic vascular resistance, which might mediate its BP-lowering effect. The similar preventive effect of a high-fiber diet and acetate supplementation on BP elevation, cardiac hypertrophy, and cardiorenal fibrosis was demonstrated in the DOCA model as well [41]. Acetate also induced the downregulation of renin-angiotensin system components in the kidney [41]. On the other hand, Olfr78, one of the SCFA receptors, might promote BP elevation, possibly through renin secretion [68]. 

As indicated above, SCFAs engage in blood pressure control, possibly through GPRs. SCFAs’ effects on BP are not uniform and depend on the receptors on which it acts. This mechanism may work to regulate BP as appropriate depending on the pathological condition.

### 4.3. SCFA’s Effect on Cardiac Inflammatory Response

As mentioned above (see Section 2.1), SCFAs show a protective effect on cardiac injury due to pressure overload and/or ischemia through the inhibitory effect of HDACs that are accompanied by the inhibition of NF-κB activity [26,27,28,29]. Besides this mechanism, SCFAs might inhibit cardiac inflammatory response in the pathologic condition, possibly through the regulation of regulatory T-cell (Treg). Treatment with resistant starches and acetate significantly increased the number of Treg cells in the spleen [37]. Consistent with this, the addition of resistant starches resulted in the regulated gene expression implicated in Treg function, which is concordant with differential DNA methylation [37]. Propionate significantly attenuated cardiac hypertrophy, fibrosis, vascular dysfunction, and hypertension in angiotensin II-infused mouse model. The cardioprotective effects of propionate were abrogated in Treg-depleted angiotensin II-infused mice, suggesting that the effect is Treg-dependent [69].

Based on the above, SCFAs may contribute to the prevention of cardiovascular disease progression from an immunological perspective by increasing the expression of Treg cells and reducing proinflammatory cytokines in the spleen.

### 4.4. SCFA as a Fuel Source

Humans utilize SCFAs as a small proportion of fuel source, as mentioned above (see Section 1). However, the recent reports implicate the increased utilization of SCFAs and its superiority as a fuel source in failing hearts. SCFAs have similar bioenergetic properties to ketone bodies and appear to be even more efficient fuels [70]. Ketones and SCFAs cross the mitochondrial membrane through free diffusion, which is efficient since it does not need any transport-mediated mechanism [71]. Although elevated ketone oxidation is observed in failing hearts as an alternate carbon source for oxidative ATP generation, SCFAs are shown to be more readily oxidized than ketones in isolated rat hearts after TAC surgery [72]. The failing hearts of rats (TAC) and humans (non-ischemic cardiomyopathy) have increased levels of ACSM3 (acyl co-enzyme A synthetase medium-chain family member 3) enzyme, which is a mitochondrial enzyme normally expressed at low levels in the heart that oxidizes SCFAs, including butyrate within the mitochondria [72]. The metabolome analysis of human plasma samples from artery and coronary sinus demonstrated equal increases in ketone and acetate metabolism in proportion to their circulating levels in HF [73]. Ischemic myocardium was also shown to preferably oxidize SCFA rather than glucose or long-chain fatty acid (LCFA) during ischemia that was induced by experimental coronary artery constriction [71]. Thus, SCFAs may have advantageous properties as a fuel source in HF.

### 4.5. SCFAs’ Effect on Gut

Patients with HF exhibit signs of intestinal dysfunction, including morphological changes such as increased gut wall thickness, indicating edema, and higher permeability [74]. In TAC-induced pressure overload model mice, the disruption of the gut epithelium was demonstrated [51]. These pathological changes might cause detrimental consequences such as bacterial/endotoxin translocation, nutritional malabsorption, and/or dysbiosis, leading to decreased SCFAs production [51], which in turn results in the further development of HF pathology [2] (Figure 1).

SCFAs, especially butyrate, show a protective effect on the pathological changes of the gut. The supplementation of SCFAs was shown to ameliorate mesenteric ischemia reperfusion injury in rat [75]. As mentioned above, butyrate shows greater uptake in the colonic epithelium compared to the other SCFAs [11] and is the main energy source for epithelial cells in colon-regulating epithelial cell function. It also has an immunomodulatory impact on host innate and adaptive immunity [14]. Acetate was shown to exert anti-inflammatory effects on the gut in colitis, possibly through GPCR activation [76]. Butyrate was shown to reduce lipopolysaccharide (LPS)-induced injury intestinal barrier integrity and tight junction permeability in a dose-dependent manner in vitro [77]. Moreover, SCFAs, especially butyrate, might exert such protective effects through increased lipoxygenase expression, which was mimicked by HDAC inhibitor [78]. Previous investigations using an in vivo chemical-induced colitis model demonstrated that butyrate improved mucosal integrity and ameliorated intestinal permeability, possibly through an anti-inflammatory effect on the colon mucosa [79]. Butyrate, especially, decreases gut CD4 T-cell activation, proliferation, and inflammatory cytokine production more potently than other SCFAs, likely through butyrate’s ability to increase histone acetylation, and potentially through GPR43 signaling [80]. 

Butyrate has anti-inflammatory effects on macrophage in the colon through the inhibition of HDACs, not GPR-mediated signaling [81]. SCFAs might promote gut integrity by maintaining symbiosis. Indeed, by lowering the luminal pH, SCFAs can directly promote the growth of symbionts, and on the other hand, inhibit the growth of pathobionts [8]. A high consumption of fiber modified the gut microbiota populations and increased the abundance of acetate-producing bacteria in DOCA hypertensive mice and even in sham mice. Both fiber and acetate decreased gut dysbiosis, represented by the ratio of Firmicutes to Bacteroidetes, and increased the prevalence of *Bacteroides acidifaciens* [41]. Similarly, fiber intake was shown to induce a significant shift in the gut microbiome in a mouse genetic model of dilated cardiomyopathy [82]. Butyrate treatment significantly altered microbiome composition and restored hypoxia in the gut epithelia and normalized intestinal permeability in angiotensin II-induced hypertensive mice [43]. Along with this, SCFAs also significantly enhanced the mRNA level expression of tight junction protein (Tjp1) and suppressed the expression of the proinflammatory cytokines and markers of fibrosis [37]. 

However, the mechanistic link between these changes in the gut microbiota community and the protective effect on gut pathology remains unexplained and needs further investigation.

### 4.6. SCFAs’ Effect on Adipose Tissue and Adipokines

Obesity is one of the most frequently encountered medical problems, which induces a variety of complications. Obesity is also related to abnormal cardiac structure and function. A reduction in visceral adipose tissue after bariatric surgery was shown to be correlated with the reduction in LV mass [83]. Adipose tissue has been considered as an endocrine organ secreting bioactive adipokines, including adiponectin and leptin, etc., which have a significant role in regulating glucose and lipid metabolism. The altered function of adipose tissue in obesity results in the dysregulated production of adipokines, which in turn favors inflammation and insulin resistance [84]. Recent evidence has shown that these adipokines are involved not only in metabolism, but also cardiovascular pathology, especially obesity-related HF with preserved ejection fraction (HFpEF), which is characterized by LV hypertrophy and/or diastolic dysfunction. Leptin is suggested to be pro-inflammatory adipokine, which is a major stimulus for the production of aldosterone in obesity [84,85], leading to promoted TNF-α production and macrophages activation [86]. Leptin might be responsible for the exacerbated mineralocorticoid receptor signaling in obesity-related HFpEF [87], possibly through impaired calcium handling and relaxation in the heart [88]. On the other hand, adiponectin has anti-inflammatory properties [89,90] and modulates oxidative stress-induced autophagy [91] and cardiac remodeling [92]. A reduced level of adiponectin is a consistent feature among obese patients who have evidence of insulin resistance and/or diabetes mellitus [93]. The plasma level of adiponectin is low in patients with coronary artery disease [89,94]. A low adiponectin level is associated with the progression of LV hypertrophy accompanied by diastolic dysfunction [95] and predicted cardiovascular events in patients with mild to moderate kidney disease [96].

SCFAs such as butyrate were shown to diminish increased insulin levels, adipocyte size, and macrophage content in epididymal adipose tissue induced by a high-fat diet. Butyrate also suppressed the plasma levels of proinflammatory adipokines such as leptin [86,97,98] and restored the depressed level of adiponectin induced by a high-fat diet [97,98]. From these findings, SCFAs might have a potential of engagement in the HF pathology via the modulation of adipokines, although its experimental or clinical evidence is still insufficient.

### 4.7. SCFAs’ Effect on Kidneys

Obesity also induces renal damages. Although its mechanisms are not fully understood, adipose tissue around the kidneys was shown to be an independent predictor of renal dysfunction, which implies a mechanistic link between adipocytes and renal damages [99]. Adipocytokine leptin has proven to be a factor that contributes to renal disease, mainly through mechanisms of the TGF-β pathway [100]. Insulin resistance causes glomerular hyperfiltration, resulting in microalbuminuria [93]. Moreover, insulin stimulates the synthesis of IGF-1 and IGF-2, both promoting glomerular hypertrophy [100]. SCFAs might be protective against these pathways, possibly through the suppression of leptin [86,97,98] and restoration of adiponectin [97,98].

In the murine model of folic acid-induced acute kidney injury (AKI), treatment with acetate, butyrate, or propionate all attenuated AKI development. A high-fiber diet also attenuated AKI development, but not in the absence of GPR41 or GPR109A. HDAC activity was inhibited in the kidneys of high-fiber fed or SCFA-treated mice [26]. Butyrate treatments significantly attenuated AKI induced by renal ischemia reperfusion injury [44].

In the CKD model induced by nephrectomy, the animals suffered a significant loss of intestinal tight junction proteins, colonic mucin, and an increase in circulating LPS, suggesting a leaky gut phenomenon. Butyrate treatment attenuated these changes and improved renal function [101]. Acetate markedly reduced renal fibrosis in DOCA hypertensive mice. Interestingly, a high-fiber diet did not show such a protective effect. The mechanisms by which fiber and acetate exert a beneficial effect are not entirely the same. Fiber upregulated GPR ligand-binding and immunoglobulin A production for the reinforcement of the intestinal immune network, whereas acetate downregulated the metabolism of butyrate and propionate in the kidney, potentially augmenting the effect of these SCFAs [41].

Thus, SFCAs act protectively against acute and chronic renal injury through a different pathway than dietary fiber supplementation.

### 4.8. Negative Effect of SCFAs

Butyrate significantly enhanced the high phosphate-induced calcification and osteogenic transition of vascular smooth muscle cells (VSMC) in vitro, whereas acetate and propionate had no effects. HDAC inhibitor trichostatin A showed inductive effects on calcification and the osteogenic transition of VSMCs, similar to butyrate NF-κB inhibitor significantly attenuated the butyrate-induced calcification. Knockdown of GPR41, but not GPR109A-attenuated butyrate-induced VSMC calcification [102].

## 5. Clinical Perspectives

As mentioned above, the evidence of SCFAs’ contribution to the pathology of HF is limited. From this viewpoint, it is still challenging to discuss the clinical perspectives regarding the role of SCFAs as a therapeutic target in humans. We need to keep in mind that a gap of knowledge persists in this research area to translate experimental findings into clinical practice.

Based on the above-mentioned findings, several potential strategies can be proposed in order to benefit from SCFAs’ protective effect against HF. A high-fiber diet is easily implemented, universally available, and can be the most cost-effective strategy. SCFAs pro-drugs or probiotics that are rich in bacteria-producing SCFAs are also a candidate for practical application if they can overcome the issues of tolerability in humans.

## 6. Conclusions

In conclusion, SCFAs are key molecules mediating the beneficial effect of microbiota on the pathology of HF via various mechanisms including HDAC inhibition, GPR-mediated effect, and efficient energy source. On the other hand, SCFAs and their signaling can be regulated by cardiovascular disease, including HF. Taken together, SCFAs might have a mutual interaction with the pathology of HF and play a significant role as the facilitator of HF. To better understand the mechanisms of these mutual relationship and its clinical application, further investigation is warranted. SCFAs have potential as novel therapeutics, leading to future personalized medicine that is implemented by characterization of the gut microbiome in HF patients.

## Figures and Tables

**Figure 1 jpm-12-01805-f001:**
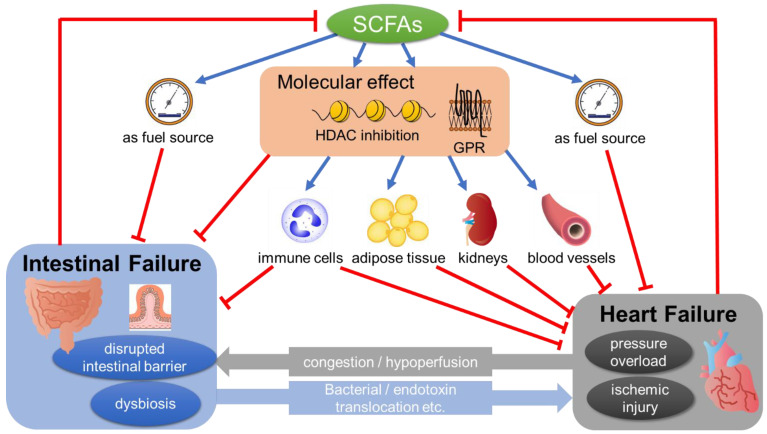
SCFAs mediating the pathology of the gut–heart axis. SCFAs show a multifaceted effect, which directly or indirectly affects the pathology of heart failure (see Section 4 in the text). Heart failure can induce pathological changes in the gut such as dysbiosis and/or the disrupted intestinal barrier, possibly through congestion and/or hypoperfusion. These changes, in turn, have a detrimental effect on the pathology of HF through bacterial/endotoxin translocation. SCFA, short-chain fatty acid; GPR, G-protein-coupled receptors; HDAC, histone deacetylases.

## Data Availability

The study did not report any data.

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
