# Peer review of "Short-Chain Fatty Acids in Gut–Heart Axis: Their Role in the Pathology of Heart Failure"

_jpm, 2022, doi:10.3390/jpm12111805_

Round 1

Reviewer 1 Report

It is an interesting paper regarding the role of short chain fatty acids in the pathophysiology of heart failure. This role is related to the heart-gut axis. Overall, the subject is of clinical interest and is well presented.

Author Response

- Reviewer 1

It is an interesting paper regarding the role of short chain fatty acids in the pathophysiology of heart failure. This role is related to the heart-gut axis. Overall, the subject is of clinical interest and is well presented.

Response: We appreciate the Reviewer for his/her review and thoughtful comments. 

Reviewer 2 Report

The manuscript represent a comprehensive review of a novel and stimulating approach.

However, its present form may result into misenterpretation if not misleading conclusions.

Authors focus mainly on findings from basic science and/or animal models.

No relevant evidence derives from clinical studies.

This should be acknowledged from the beginning. Accordingly, paragraph 5 should not result in not grounded conclusion or recommendations

It may be organized as priorities ingap of knowledge and/or future priority in research to translate experimental findings into clinical practice

SPECIFIC COMMENTS

- it should be interesting to discuss the cross-talk between metabolic alteration and kidney (PMID: 21403882), that is so relevant for heart performance

- similarly, talking about heart failure and the neccessary role of kidney function, the novel perspectives about mineralocorticoid receptors in adipose tissue and other "unexpected" cells (macrophages, etc) (PMID: 31843490) may broaden the perspective of the reported observations

Author Response

- Reviewer 2

We thank the Reviewer for his/her thoughtful comments.  Below are our responses:

The manuscript represent a comprehensive review of a novel and stimulating approach.However, its present form may result into misenterpretation if not misleading conclusions.Authors focus mainly on findings from basic science and/or animal models. No relevant evidence derives from clinical studies. This should be acknowledged from the beginning. Accordingly, paragraph 5 should not result in not grounded conclusion or recommendations. It may be organized as priorities ingap of knowledge and/or future priority in research to translate experimental findings into clinical practice

Response: We appreciate the Reviewer for his/her important suggestion. According to the Reviewer’s suggestion, we added the comments on the very sparse clinical evidence in this research area, which is shown in lines 59-65 in Introduction section. And, in paragraph 5, we added the comments that there is still a gap of knowledge in research to translate experimental findings into clinical practice, which is added in lines 423-427. Further, the overall expression is weakened based on the recognition that the evidence is still insufficient.

 SPECIFIC COMMENTS

- it should be interesting to discuss the cross-talk between metabolic alteration and kidney (PMID: 21403882), that is so relevant for heart performance

- similarly, talking about heart failure and the neccessary role of kidney function, the novel perspectives about mineralocorticoid receptors in adipose tissue and other "unexpected" cells (macrophages, etc) (PMID: 31843490) may broaden the perspective of the reported observations

Response: We appreciate the Reviewer for his/her advice. Metabolic derangement, adipocytokines and their contribution to renal impairment can be related to the pathology of obesity related heart failure and renal damage. We created new subsection of “4.6. SCFAs’ effect on adipose tissue and adipokines” and discussed the potential protective role of SCFAs in obesity related heart failure via adipokines, citing 2 references recommended by the Reviewer. And we added the discussion on potential protective role of SCFAs in obesity induced renal damage in subsection “4.7. SCFAs’ effect on kidneys” . In the Figure, SCFAs’ effect via adipocytes was additionally depicted.

Reviewer 3 Report

Thank you for allowing me to review this very interesting manuscript titled, 'Short Chain Fatty Acids in Gut-heart Axis: Their Role in the Pathology of Heart Failure'. The authors have done a commendable job in describing the role of short-chain fatty acids in the pathogenesis of heart failure. I have a few questions.

1. Do you ascribe the SCFA to the pathogenesis of heart failure, or in your opinion, are SCFAs a surrogate marker for heart failure?

2. In patients with transient low cardiac output states, would this hypothesis still hold true? 

3. What in your opinion would happen to the SCFAs should the heart failure be reversed, by means of inotropes, assist devices or heart transplantation? Is these any evidence for the same?

Author Response

- Reviewer 3

We thank the Reviewer for his/her thoughtful comments.  Below are our responses:

Thank you for allowing me to review this very interesting manuscript titled, 'Short Chain Fatty Acids in Gut-heart Axis: Their Role in the Pathology of Heart Failure'. The authors have done a commendable job in describing the role of short-chain fatty acids in the pathogenesis of heart failure. I have a few questions.

  1. Do you ascribe the SCFA to the pathogenesis of heart failure, or in your opinion, are SCFAs a surrogate marker for heart failure?

Response: As stated in the text and Figure 1, there can be a mutual interaction between SCFA and heart failure. Although the evidence is still limited, we speculate SCFAs might play a significant role as the facilitator of heart failure. We added the comments in Lines 437-439 in the Conclusions.

  1. In patients with transient low cardiac output states, would this hypothesis still hold true? 

Response: Although clinical evidence is sparse, it has been reported that the composition of intestinal microbiota changed in animal models of intestinal ischemia and reperfusion. These changes might affect SCFA-producing bacteria, although there have been no direct evidence for now. This description was added in lines 34-39 in the Introduction. We also added the description on the potential relationship between microbiota and heart failure in the Introduction.

  1. What in your opinion would happen to the SCFAs should the heart failure be reversed, by means of inotropes, assist devices or heart transplantation? Is these any evidence for the same?

Response: To our knowledge, there have been no evidence supporting restoration of SCFA concentration by means of inotropes, assist devices or heart transplantation. Recent study demonstrated that, contrary to expectations, microbiota-derived metabolite trimethylamine N-oxide (TMAO), which is elevated in HF and linked to poor prognosis, showed decrease early following left ventricular assist device (LVAD) or heart transplantation (HT) but re-increased slowly to preintervention levels. Lipopolysaccharide, a biomarker of endotoxemia was also shown to remain elevated even after LVAD or HT. Thus, it is hard to anticipate how the levels of SCFAs will change after LVAD or HT and it needs to be investigated in future. We added the description in lines 188-196 in section 3.

Round 2

Reviewer 2 Report

The manuscript has largely been improved

The novel section on adipokines and obesity is interesting and may open several considerations. At least, it should be discussed the impact of "body size" on blood pressure measurement accuracy ( see DOI 10.1097/HJH.0000000000002246)...

....given the relevance of BP values to start, discontinue, change treatment in HF patients and theri eligibility for allocation to innovative drugs

Author Response

- Reviewer 2

We thank the Reviewer for his/her thoughtful comments.  Below are our responses:

The manuscript has largely been improved

The novel section on adipokines and obesity is interesting and may open several considerations. At least, it should be discussed the impact of "body size" on blood pressure measurement accuracy ( see DOI 10.1097/HJH.0000000000002246)...

....given the relevance of BP values to start, discontinue, change treatment in HF patients and theri eligibility for allocation to innovative drugs

Response: According to the Reviewer’s comments, we added the description on the importance of BP evaluation in the management of HF, which is in lines 225-228.

Reviewer 3 Report

The authors have satisfactorily addressed my concerns. Commendable effort.

Author Response

We appreciate the Reviewer for his/her thoughtful comments to improve our paper.